# Migration Pattern, Habitat Use, and Conservation Status of the Eastern Common Crane (*Grus grus lilfordi*) from Eastern Mongolia

**DOI:** 10.3390/ani13142287

**Published:** 2023-07-12

**Authors:** Baasansuren Erdenechimeg, Gankhuyag Purev-Ochir, Amarkhuu Gungaa, Oyunchimeg Terbish, Yajie Zhao, Yumin Guo

**Affiliations:** 1School of Ecology and Nature Conservation, Beijing Forestry University, Beijing 100083, China; baaskaa.1227@gmail.com (B.E.); pgankhuyag@gmail.com (G.P.-O.); 2Mongolian Bird Conservation Center, Ulaanbaatar 14200, Mongolia; gamarkhuu@gmail.com; 3Eastern Mongolian Protected Areas Administration, Choibalsan 21060, Mongolia; oyunaaterbish@gmail.com; 4Shandong Yellow River Delta National Nature Reserve Management Committee, Dongying 257091, China; zhaoyajie-2005@163.com; 5Technology Innovation Center for Ocean Telemetry, Ministry of Natural Resources, Qingdao 266061, China

**Keywords:** Eastern common crane, migration pattern, stopover site, satellite tracking, habitat use, conservation gap

## Abstract

**Simple Summary:**

Studies on the migration patterns, habitat use, and conservation of the Eastern common crane *Grus grus lilfordi* in East Asia are insufficient. Most of the summering, breeding, wintering, and stopover sites are located outside the current protected areas boundary, so it is necessary to pay attention to these areas for the future protection of this subspecies.

**Abstract:**

Studies on the subspecies Eastern common crane *Grus grus lilfordi* are still scarce, especially in Southeastern Siberia, the far east of Russia, Eastern Mongolia, and Northeastern China. This study explores the migration pattern, habitat use, and conservation status of the Eastern common crane. Using GPS/GSM tracking data, 36 complete migrations of 11 individuals were obtained from 2017 to 2021. The cranes migrated an average of 1581.5 km (±476.5 SD) in autumn and 1446.5 (±742.8 SD) in spring between their breeding site in Eastern Mongolia and the following wintering sites: the Xar Moron River, Chifeng; the Bohai Bay; the Yellow River Delta; Tangshan, Hebei; and Tianjin. During the autumn and spring migrations, the cranes used three critical stopover sites. The subspecies spent 60.3% of their time in rangeland, 18.1% in cropland, and 14.2% in water. The tracking data determined that, of the areas used by cranes, 97–98% of the summering sites were in Russia, 96% of the breeding sites were in Mongolia, and over 70% of the stopover sites and 90% of the wintering sites in China lay outside the current protected area boundaries. Consequently, establishing and expanding protected areas in summering, breeding, stopover, and wintering sites should be a central component of future conservation strategies.

## 1. Introduction

In the early 1990s, with the continuous development of science and technology, the research methods for studying bird migration also made significant progress, allowing researchers to better understand the migratory behavior of birds. In recent years, research on the long-distance migration of birds using satellite tracking technology has intensified [1,2]. Understanding the migration duration, migration routes, stopover sites, and other information about migratory birds can provide a scientific basis for bird conservation and protected area management [3,4].

The common crane *Grus grus* is assessed as a least concern (LC) species in the IUCN Red List [5]. Currently, four subspecies of the common crane have been recognized: the Western common crane *G. g. grus* in Europe; the Transcaucasian subspecies *G. g. archibaldi* in the Inner Caucasus/borderland of Turkey, Georgia, and Iran [6,7]; the Tibetan subspecies *G. g. korelovi* in Tibet, Kyrgyzstan, and Kazakhstan [8]; and the Eastern common crane *G. g. lilfordi* in Asia from the Ural Mountains eastward to Northeastern Siberia [9,10]. The Western and Eastern common cranes are separated by the Ural Mountains [11]. The population size of the eastern subspecies is estimated at 125,000–130,000 individuals [12], of which more than 100,000 migrate from Western Siberia and Kazakhstan to the wintering grounds in India and Central Asia in the Amudarya River Valley [13,14,15]. Central/Eastern Siberia, Mongolia, and Northeastern China provide the main breeding sites of the Eastern common crane, with the population estimated to be about 12,000–20,000 individuals [12,16,17]. There are also several major wintering sites such as the middle and lower reaches of the Yangtze River, the Yellow River Delta wetlands, Poyang Lake, the Shengjin Lake wetlands, the Mengjin Yellow River tidal flat wetlands, the Yancheng coastal wetlands, and the Beijing Wild Duck Lake wetlands [18,19,20].

In the last century, due to population growth, social and economic development, construction, and the use of wetlands, the habitat of the Eastern common crane has changed significantly and suitable habitat has been lost [17].

In this study, we aim to identify the migration patterns, habitat use, and conservation gaps of the Eastern common crane from Eastern Mongolia through satellite tracking and provide the basic information necessary for future conservation of this subspecies.

## 2. Materials and Methods

### 2.1. Satellite Tracking

We tracked 11 Eastern common cranes (3 adults and 8 hatch-year juveniles) along the Ulz River basin and its surrounding lakes in Eastern Mongolia from May to August in 2017 (*n* = 3, all juveniles), 2018 (*n* = 3, all adults), and 2019 (*n* = 5, all juveniles). Adults were successfully caught during molting, whereas juveniles were safely caught before their wings had fully grown. Each crane’s head was hooded after capture using clothing to reduce stress. Solar-powered GPS-GSM satellite transmitters (model HQLG4021S; Global Messenger Technology Company, Changsha, China) were fitted to each crane’s right leg, and on the left leg was fitted a green ring (with three numerals) with an inner diameter from 18 to 20 mm. All transmitters and rings were fitted in less than two to three minutes. After careful observation to ensure that the birds were not suffering from capture stress associated with capture and handling, birds were released at the capture site 5–10 min after transmitters were deployed. The transmitter and ring weighed a total of 44 g and were 0.8–1.6% of each crane’s body weight (Appendix A), which is in accordance with the weight regulation of less than 3–5% of the bird’s mass [21]. The satellite transmitter recorded GPS coordinates, speed, altitude, temperature, precision, and battery voltage hourly.

### 2.2. Migration Analysis

The start and end points of migration were outlined by the tracking data for which individuals departed from and arrived at summering and wintering sites, respectively [22].

The date of the first position when a bird left a wintering or breeding site was used to determine the departure date [23]. Based on the methods described by Wang et al. [22] to qualify for “non-flight” status after a period of flight, the arrival date was defined as the date when a bird was determined to have arrived at a wintering or breeding site.

The average departure and arrival date of autumn and spring migration was calculated using data from 36 full migration trips collected from a total of 11 individuals.

The migration duration was calculated as the time a bird took to migrate (including stopovers) between its last summering location and its first wintering location (autumn migration) or between its last wintering location and its first summering location (spring migration) [24]. The total travel distance between all adjacent GPS locations that are designated as being in “flight” status during migration is known as the “migration distance” [24,25].

The migration speed was calculated by dividing the total migration distance (in kilometers) by the total migration duration (in days) [26]. The migration distance divided by the total travel days yielded the daily travel speed [26].

The travel days are calculated as the total duration of migration (days) minus stopover days. The sum of all days spent at all stopover sites during each migration season was used to define stopover duration [26]. Stopover sites were identified as places where birds remained stationary during the migration for more than two days [25]. Sites where the cranes stayed for more than 14 days were determined to be “critical stopover sites” [27,28].

The differences between different seasons among crane migration routes and between different strategies (flying speed) used by cranes performing autumn and spring migration were examined using Mann–Whitney U tests in R 4.2.3. The data are shown as the mean ± SD.

The migration routes and KDE maps were produced using ArcGIS 10.8, and the geometry function was used to determine the cumulative migration distance. The density of cumulative bird use in two dimensions was described using kernel density estimation (KDE). After interpolation, the 11 tracked cranes generated 109,224 locations that were used to estimate the kernel density. This method was used to identify crane migration corridors during their annual migratory cycle.

Flight speed was calculated by filtering data of cranes that flew over 10 km/h during spring and autumn migration.

Flight altitude data (flight speed above 10 km/h) were used to determine the flight altitude of the cranes. Digital Elevation Data (DEM) originated from USGS Earth Explorer (https://earthexplorer.usgs.gov; accessed on 26 March 2023). Based on the DEM with a spatial resolution of 30 m, the flight altitude maps were created in ArcGIS 10.8 software.

The Wilcoxon test was used to test the differences between the flight altitude of daytime and night and the differences between flight altitude during the crossing of the Khingan mountain range and other places during the migration.

### 2.3. Habitat Use and Conservation Gaps

Using the data management tools of ArcGIS 10.8, the home ranges (KDE 95%) of the tracked individuals in critical sites (summering, breeding, stopover, and wintering) were determined. For comparative purposes, the minimum convex polygon (MCP 95%) was also determined.

We used the Esri land cover data set (with an accuracy of 10 m × 10 m) created by Esri, Microsoft, and Impact Observatory to assess the habitat use of the Eastern common crane [29]. Nine land-cover categorization types are included in the Esri land-cover map, which uses Sentinel-2 satellite imagery with a 10 m resolution. These types of land cover include clouds (for areas with no data), water, trees, flooded vegetation, crops, built area, bare ground, snow/ice, and rangeland. Open areas covered in homogenous grasses with little to no taller vegetation present fall under the rangeland category. By overlaying the crane positions on the mapped coverage of the land-cover classification classes described above in ArcGIS 10.8, the proportion of habitat occupied by tagged cranes during the day and night at each summer, breeding, stopover, and wintering site was calculated.

To determine the conservation gaps at the summering, breeding, stopover, and wintering sites of the Eastern common crane, we overlaid the home range (MCP 95%) of the cranes with the protected area boundaries (WDPA 2023; accessed at protectedplanet.net; accessed on 16 April 2023) in ArcGIS 10.8.

## 3. Results

### 3.1. Migration Patterns

We accrued 109,224 bird locations from 11 individual Eastern common cranes, covering 36 full migration trips (20 autumn migration trips and 16 spring migration trips) from July 2017 to November 2021 (Figure 1 and Appendix A). The tracked cranes migrated an average of 1581.5 km (±476.5 SD) in autumn and 1446.5 km (±742.8 SD) in spring (Table 1) between their breeding site in Eastern Mongolia and the wintering sites in China (shown systematically in Appendix A).

Cranes spent an average of 150 days (±10) in breeding sites in Eastern Mongolia, departing on average on 8 October (±44 days), and migrating for an average of 49.6 days (±32.8). The cranes arrived at the following wintering sites: the Xar Moron River in Chifeng, Inner Mongolia; the Bohai Bay (Jinzhou, Liaoning province); the Yellow River Delta (Dongying, Shandong province); the Luan River, Tangshan, Hebei province; and the coastal area of Tianjin. They arrived from late August to early December (30 October ± 61), where they spent an average of 180 days (±30). The spring migration on average started on 7 April (±43 days), with migration occurring for 28.5 days (±22.4), and an average date of arrival being 2 May (±27 days) (Table 1).

There was no significant difference between autumn (2.3 days ± 0.7) versus spring (3.3 days ± 2.7) travel duration, migration speed (*p* = 0.08), and migration distance (*p* = 0.61), but the autumn migration duration (49.6 days ± 32.8) was longer than spring migration duration (28.5 days ± 22.4) (Figure 2).

During autumn migration, the flight speed averaged 53.9 ± 22.8 km/h, ranging from 10.2 to 144.7 km/h. The earliest time when individuals started to migrate in a day was between 06:00 and 08:00, and the latest time was between 21:00 and 23:00. During spring migration, the flight speed averaged 48.5 ± 21.48 km/h, ranging from 10.5 to 145.3 km/h. The earliest time when individuals start to migrate in a day was 06:00, and the latest time was between 23:00 and 00:00. Although the nocturnal flight speed was relatively slow, there was a significant difference between the hour of the day and the flight speed for autumn migration and spring migration (*p* < 0.05) (Figure 3).

The flight speed below shows that the tracked cranes mostly migrated during the daytime, but three individuals (CC061, CC116, and CC201) migrated at night.

The average altitude at which the cranes flew was 639 m; however, they tended to fly higher when crossing the Khingan mountain range (mean = 688 m) compared with elsewhere (mean = 553 m) (Figure 4).

The cranes occasionally flew higher, in some places reaching altitudes as high as 2371 m, and in the Khingan mountain range as high as 2859 m. There was a difference in flight altitude between crossing Khingan mountain and other places (Wilcoxon test, w = 276, 478; *p* < 0.05). In addition, flight altitudes were different between day (mean = 650 m) and night (mean = 494 m) flights (Wilcoxon test, w = 66,820; *p* < 0.05).

The duration for which data was collected for each individual varied, and for some individuals, data collection was interrupted and halted, possibly due to tracker malfunction. However, of the birds that finished a full migration, two cranes (CC279, CC281) did not return to their parents’ nesting territory (precise point of capture), and three cranes (CC116, CC119, and CC277) returned to their parents’ nesting territory in the second and third years. We observed crane CC277 in Turgenii Tsagaan Lake (captured site of CC277) in June 2021, and witnessed that CC277 was already paired and summering there.

### 3.2. Stopover Sites

In total, we identified three important stopover sites along the migration route. The Khalkh Gol (river valley) in far Eastern Mongolia is the first critical stopover site during the autumn and spring migrations. Eight cranes used the Khalkh Gol as a stopover site averaging 38 days (±17) there in autumn, and four cranes spent an average of 9 days (±7) there in spring (Figure 5).

For some cranes (*n* = 2 in autumn, *n* = 5 in spring), Xilin Gol in Inner Mongolia was also a critical stopover site in autumn (24 days) and spring (13 days ± 12).

The Xar Moron River in Chifeng was also a critical stopover site during the autumn (42 days) and spring (44 days) migration for Eastern common cranes. The KDE map (Figure 6) shows that the Xar Moron River was not only a critical stopover site but also an important wintering site and summering site for this subspecies. The following tracked cranes wintered in the Xar Moron River: CC185 in 2017; CC119 in 2018 and 2019; CC210 in 2018; CC281 in 2019 and 2020; and CC277 in 2019 to 2021. Crane CC279 summered in the Xar Moron River from 2020 to 2021 (Figure 1).

There were no statistical differences in the number of stopovers in the autumn and spring migrations (*p* = 0.41). However, the autumn stopover duration (43.6 days ± 31.1) was longer than the spring stopover duration (25.3 days ± 21.9) (Table 1).

### 3.3. Habitat Use and Conservation Gaps

We determined that tracked Eastern common cranes spent 60.3% of their time in rangeland, 18.1% in cropland, and 14.2% in water (habitat types with very low percentages are systematically shown in Table 2). In summering sites, the tracked cranes mostly used rangeland (60.2%) and water (30.9%). In terms of breeding sites, rangeland (84.5%) was an important habitat (Figure 7). For rangeland (59.8%) and cropland (25.3%) habitats, use was significantly more in stopover sites, but rangeland (12.5%) was less used in winter. In all wintering sites, tracked cranes mostly used water (47.8%) and cropland (31.8%).

The tracking data determined that two tracked cranes summered in Zabaykalsky and Aginskiy Rayon in Russia. The total summering site home range (MCP 95%) size is estimated to be 867 km^2^, of which 2–3% is protected and 97–98% has no protection. The breeding site (Eastern Mongolia) home range is estimated to be (MCP 95%) 16,945 km^2^ and (KDE 95%) 3841 km^2^, of which 96% is outside of the protected area (Figure 8).

Of the three stopover sites identified in the migration route, 30% of the Tashgain Tavan Lakes Nature Reserve, including its cropland, and the Khalkh Gol valley (MCP 95%—1426 km^2^) in far Eastern Mongolia coincided with the protected area boundary. In this stopover site, we recorded a total of 2988 common cranes congregating during the autumn migration in September 2022. This number of individuals accounts for 14.9–24.9% of the population of the Eastern common crane *G. g. lilfordi*.

The other two stopover sites (home range size estimated to be 1395 km^2^ in Xilin Gol and 6550 km^2^ in Xar Moron River, Chifeng) have no protection. The wintering home range is estimated to be a total of 2537 km^2^ (Liaoning—439 km^2^; Tangshan, Hebei—170 km^2^; Tianjin—1195 km^2^ and Shandong—733 km^2^), of which 10% is protected in Tianjin and 70% is protected in Shandong, while 100% has no protection in Liaoning and Tangshan (shown systematically in Figure 8).

## 4. Discussion

### 4.1. Migration Patterns and Stopover Sites of the Eastern Common Crane

According to the results of our study, the migration distance of the Eastern common crane was 1581.5 km in autumn and 1446.5 km in spring. This is relatively short compared to the average migration distance of the Western common crane *G. g. grus* (2400–5100 km) [30] but comparatively longer than the migration distances of the Transcaucasian common crane *G. g. archibaldi* (600–1000 km) and the Tibetan common crane *G. g. korelovi* (400–500 km) [31,32].

The duration of the autumn migration (49.6 ± 32.8) is longer than the duration of the spring migration (28.5 ± 22.4), which is consistent with the research conducted on the Western common crane in Europe [30,33]. Additionally, compared to other species of cranes, we found that the migration pattern of the Eastern common crane is most similar to that of the Siberian crane (*Leucogeranus leucogeranus*), which uses the same flyway in east Asia [28]. However, it is different from the migration pattern of White-naped cranes (*Antigone vipio*), which also migrate from Eastern Mongolia [34]. The autumn migration duration of the Eastern common cranes was longer than the autumn migration duration of White-naped cranes that migrated from Mongolia. This can be explained by the difference in the number of stopover sites of the cranes, as White-naped cranes use only one stopover site for each of the autumn and spring migration periods. We found that during the autumn migration, tracked Eastern common cranes used the croplands and wetlands around the Tashgain Tavan Lakes Nature Reserve (including its cropland areas) and the Khalkh River valley in far Eastern Mongolia for a long time before crossing the Khingan mountain. After crossing the Khingan mountain, the cranes rested in the croplands of the Xilin Gol in Inner Mongolia and the Xar Moron River in Chifeng, acquiring energy to travel to wintering sites. Birds expend more energy when crossing ecological barriers such as deserts, oceans, and high mountains, and rest for a long time at stopover sites to acquire enough energy to cross these geographic barriers [35]. The Khingan mountain has an average elevation of 1200 to 1300 m, with the highest peak reaching 2035 m. The flight altitude of the Eastern common crane when crossing the Khingan mountain range is higher than their flight altitude elsewhere. Consequently, it is likely that a lot of energy was consumed when crossing the Khingan mountain range. Therefore, the critical stopover sites are located at sites to the south and north of the Khingan mountain range, which provide the necessary stopover conditions for Eastern common cranes.

Liu et al. [36] determined the migration and stopover sites of wintering common cranes in the Caohai National Nature Reserve, southern China; the cranes used Zhongwei as a long-term stopover site before crossing the Qinling mountains and the Ala Shan Desert. This migration strategy has also been observed in migration studies of the common crane [30,31]. We hypothesize that the study by Liu et al. [36] covered a population of the Eastern common cranes breeding in Western Siberia, and although the distance and migration duration is not the same as that found in our study, most of the migration duration was spent at the stopover sites for rest. We also found a similar strategy from our tracked Eastern common cranes in Eastern Mongolia.

### 4.2. Migration Strategy and Site Fidelity of the Eastern Common Crane

Our tracked cranes’ autumn migration duration and stopover duration were longer than in the spring migration. The adult cranes’ departure date was earlier than juvenile and sub-adult cranes in spring migration, perhaps because young cranes need to feed and rest more than adults. Furthermore, adults need to arrive at the breeding site as soon as possible to breed [37,38].

The Eastern common crane migration was mainly diurnal, but for some individuals, migration was nocturnal for one to two migration trips. This indicates that this subspecies can travel in both diurnal and nocturnal migrations. Although Alonso et al. [39] and Postelnykh et al. [40] determined that common cranes’ migration is diurnal, Ojaste et al. [30] and Swanberg [41] pointed out that common cranes occasionally make uninterrupted flights of more than 36 h, which is insufficient to define common cranes’ migration as primarily diurnal.

We found that five tracked individuals (CC116, CC277, CC119, CC279, CC281) in Eastern Mongolia for which we have more than one year of tracking data showed strong site fidelity during both wintering and breeding seasons. This site fidelity is also observed in the Western Eurasian crane *Grus grus grus* [42], indicating that the common crane had a high site fidelity regardless of population and age.

The success requires fitting several tens of tracking devices because just one out of ten devices, as in the present study, provides three years of data or longer. Therefore, to better understand the migration strategies of common cranes, it is necessary to track many more birds [40].

### 4.3. Habitat Use and Conservation Gaps of the Eastern Common Crane

In this study, we identified that water and rangeland were highly used in summering and breeding sites, representing over 90% of use. In stopover sites, the cranes mainly used cropland, rangeland, and water, while in the wintering sites, cropland and water were important feeding and roosting habitats of this subspecies. Our results show that rangeland (grass and reeds) occupied 84.6% of the breeding habitat of the Eastern common crane. According to studies by Batbayar et al. [27] and Lang et al. [34], grass occupied 89.3% of the breeding habitat of the White-naped crane in Eastern Mongolia. These results indicate that rangeland (grass and reeds) habitats in Eastern Mongolia are the most important habitats for these two species of crane. Herder family and livestock numbers are increasing along the river basins [43], and subsequent overgrazing degrades wetlands and reed habitats, which are important breeding habitats for Eastern common cranes.

Therefore, we suggest increasing the knowledge of the local people living along the rivers and surrounding lakes about nature conservation and also improving the planning and execution of conservation management of important breeding habitats along the Ulz River.

Our analysis revealed that most of the breeding (conservation gaps 96%), stopover, and wintering sites are located outside of the current protected area boundaries. During our field survey, we found that Eastern common cranes gathered during their autumn migration in the cropland of Khalkh Gol, a stopover site during their migration, have been scared and driven away by farmers through gunshots. Establishing and expanding protected areas do not ensure that farmers do not scare and drive away the common cranes. Achieving a more balanced management plan with an effective combination of multiple management practices is essential [44].

## 5. Conclusions

This study precisely quantified the migration patterns (migration routes, stopover sites, flight speed, and flight altitudes), habitat use, and conservation status of the Eastern common crane. The migration distance of the Eastern common crane was shorter than its Western subspecies *G. grus*, but the basic migration strategy was similar. The main strategy of the Eastern common crane’s autumn migration was to travel long distances in a short period, achieved by resting in a few stopover sites for several days to acquire energy for subsequent long flights. In total, we identified three critical stopover sites along the migration route: the Tashgain Tavan Lakes NR, including its cropland, and the Khalkh Gol Valley, Eastern Mongolia; Xilin Gol, Inner Mongolia; and the Xar Moron River, Chifeng. Our data clearly show that Eastern common cranes rely heavily upon rangeland, cropland, and water habitats. In addition, most of the wintering, breeding, stopover, and summering sites were outside the current protected area boundaries. Based on these results, we suggest that in order to effectively maintain this species, conservation interventions targeting crucial sites along the whole migratory path are needed.

## Figures and Tables

**Figure 1 animals-13-02287-f001:**
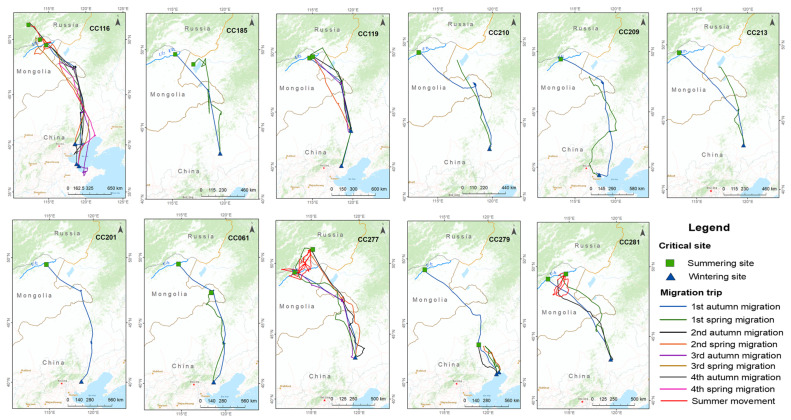
Migration routes and critical sites of each individual Eastern common crane *G. g. lilfordi*.

**Figure 2 animals-13-02287-f002:**
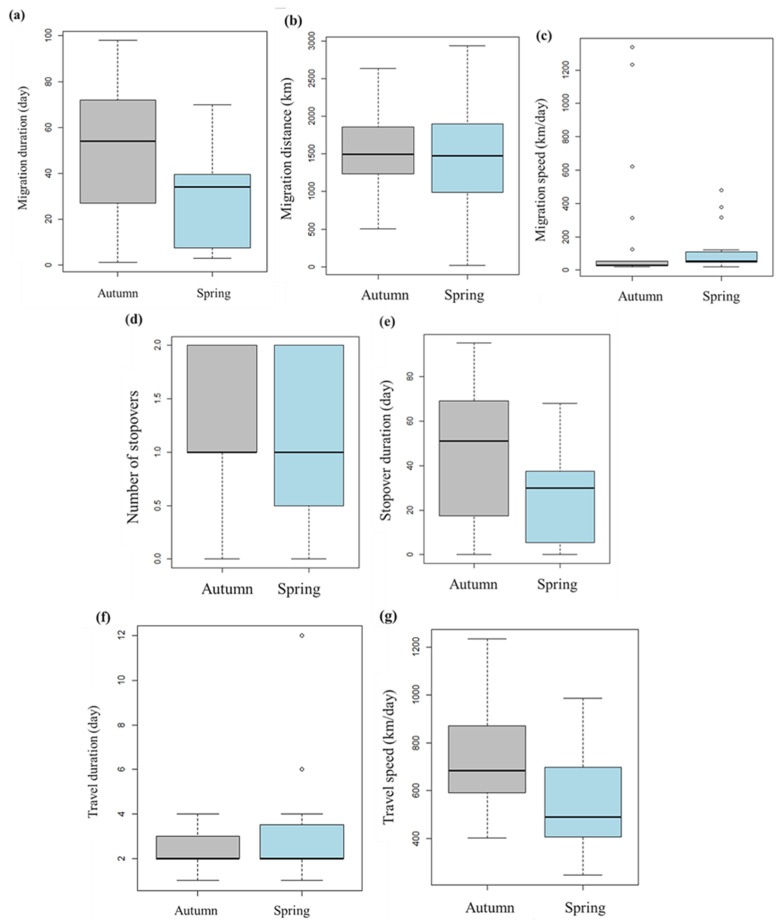
Boxplots of Eastern common crane *G. g. lilfordi* migration parameters in autumn (gray) and spring (light blue) migration. (**a**)—migration duration (day); (**b**)—migration distance; (**c**)—migration speed (km/day); (**d**)—number of stopovers; (**e**)—stopover duration (day); (**f**)—travel duration (day); (**g**)—travel speed (km/day).

**Figure 3 animals-13-02287-f003:**
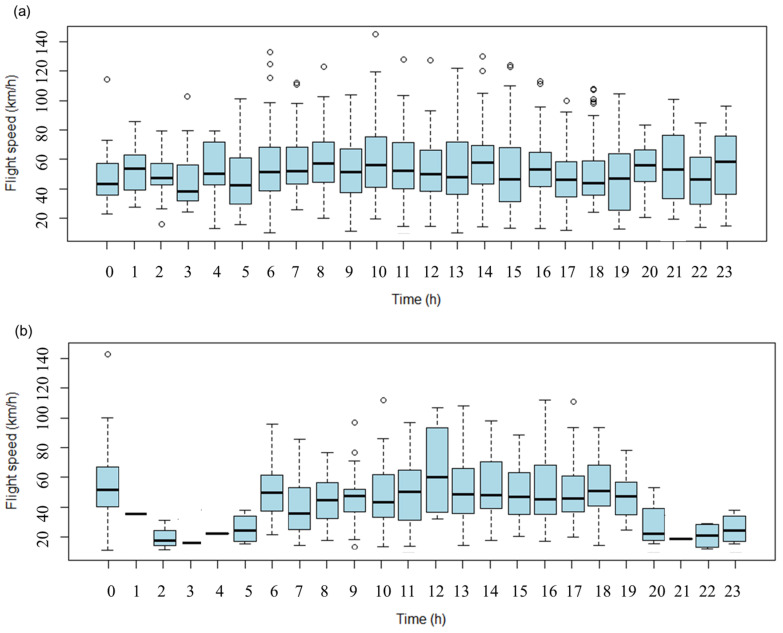
Flight speed (km/h) according to the hour of day of Eastern common cranes *G. g. lilfordi*. (**a**)—autumn migration; (**b**)—spring migration.

**Figure 4 animals-13-02287-f004:**
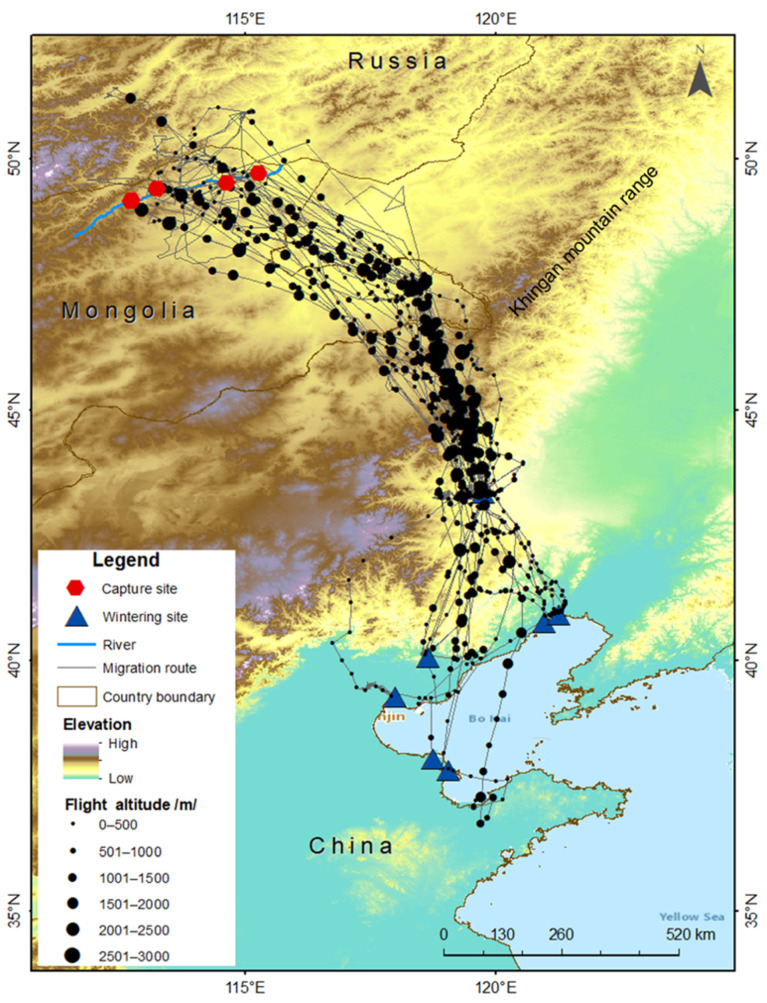
Flight altitude (m) of Eastern common cranes *G. g. lilfordi* during the migration.

**Figure 5 animals-13-02287-f005:**
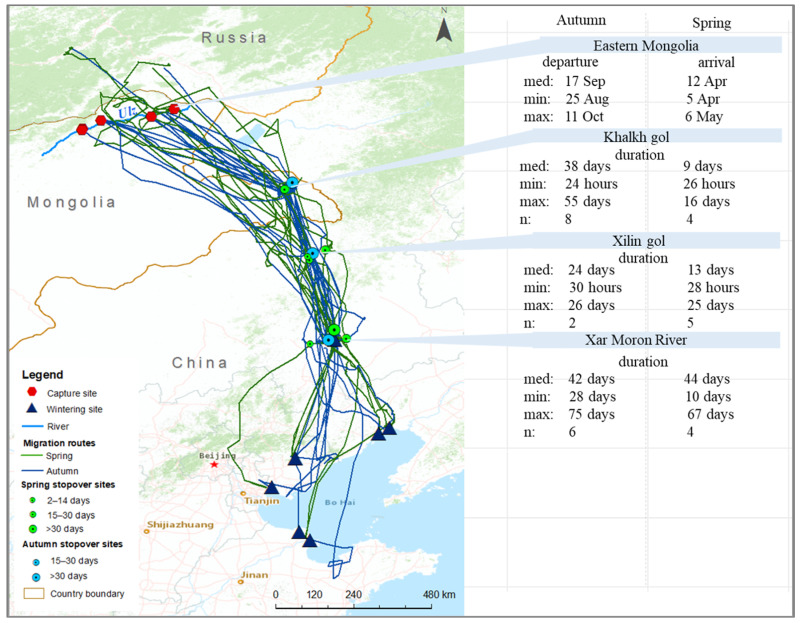
Autumn (blue) and spring (green) migration routes of Eastern Common Cranes *G. g. lilfordi* and their stopover sites (autumn—light blue, spring—light green).

**Figure 6 animals-13-02287-f006:**
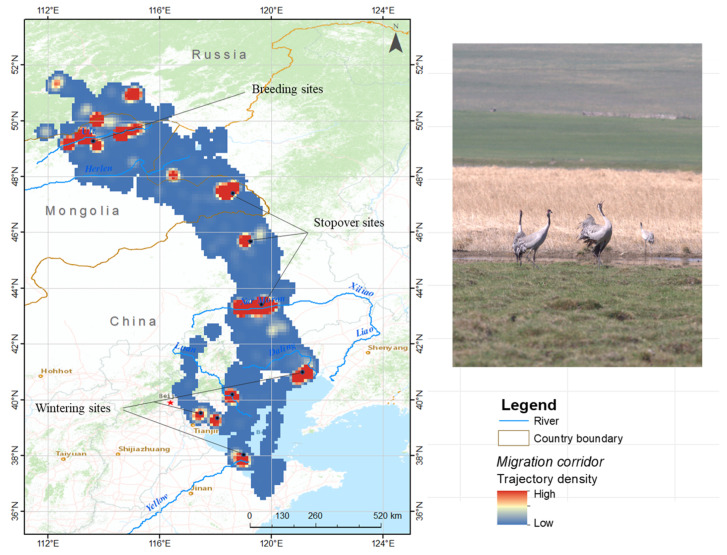
Mapping of Eastern common crane *G. g. lilfordi* migration corridors, based on Kernel Density Estimation (KDE) summarization of tracking data from 11 tracked individuals. The orange to red colors indicate the highest intensity of use, and yellow to blue the lowest.

**Figure 7 animals-13-02287-f007:**
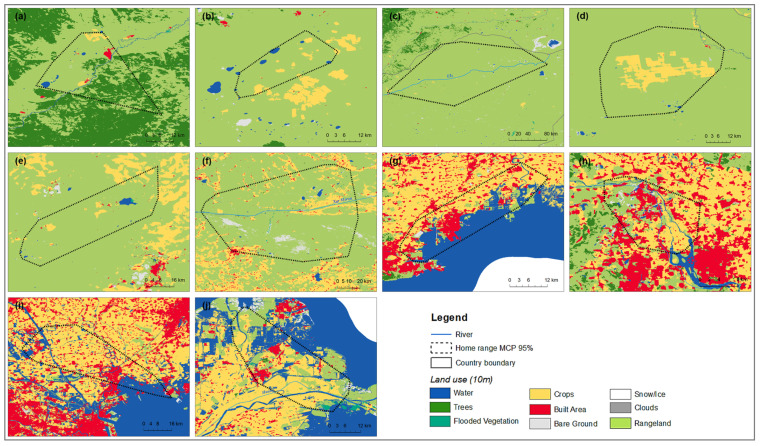
Land cover maps of Eastern common cranes *G. g. lilfordi* at summering ((**a**)—Zabaykalsky, (**b**)—Aginskiy Rayon in Russia), breeding ((**c**)—Eastern Mongolia), stopover ((**d**)—Tashgain Tavan lakes Nature Reserve including its cropland and Khalkh Gol valley in Eastern Mongolia, (**e**)—Xilin Gol in Inner Mongolia, (**f**)—Xar Moron River in Chifeng), and wintering ((**g**)—Liaoning, (**h**)—Tangshan, (**i**)—Tianjin, (**j**)—Shandong) sites with home range (MCP 95%).

**Figure 8 animals-13-02287-f008:**
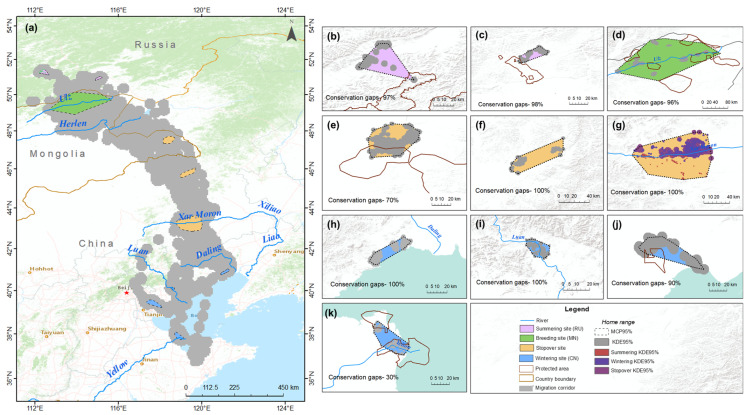
Conservation status of the Eastern common crane *G. g. lilfordi*. (**a**) Migration corridor and critical sites. ((**b**)—Zabaykalsky, (**c**)—Aginskiy Rayon) Summering (RU), ((**d**)—Eastern Mongolia) breeding (MN), ((**e**)—Tashgain Tavan lakes Nature Reserve including its cropland and the Khalkh Gol valley in Eastern Mongolia, (**f**)—Xilin gol in Inner Mongolia, (**g**)—Xar Moron River in Chifeng) stopover, ((**h**)—Liaoning, (**i**)—Tangshan, (**j**)—Tianjin, (**k**)—Shandong) wintering (CN) sites with protected areas and home ranges.

**Table 1 animals-13-02287-t001:** Statistical comparison of the autumn and spring migration parameters of Eastern common cranes *G. g. lilfordi* based on Mann–Whitney U-tests.

Parameter	Mean Value ± SD		U Test	
	Autumn Migration	Spring Migration	W	*p* Value
Departure date	8/X ± 44	7/IV ± 43	-	-
Arrival date	30/X ± 61	2/V ± 27	-	-
Migration duration (day)	49.6 ± 32.8	28.5 ± 22.4	183	0.08
Migration distance (km)	1581.5 ± 476.5	1446.5 ± 742.8	187	0.61
Migration speed (km/day)	196.6 ± 168.3	124 ± 104.3	103	0.08
Number of stopovers	1.4 ± 0.67	1.12 ± 0.81	184	0.41
Stopover duration (day)	43.6 ± 31.1	25.3 ± 21.9	194	0.07
Travel duration (day)	2.3 ± 0.7	3.3 ± 2.7	126	0.40
Travel speed (km/day)	722 ± 213	573 ± 233	187	0.06

**Table 2 animals-13-02287-t002:** Summary of habitat use of Eastern common cranes *G. g. lilfordi*.

Habitat Types	Summering (RU)	Breeding (MN)	Stopover	Wintering (CN)	Total
Water	30.9%	6.77%	7.7%	47.8%	14.2%
Trees	0.03%	0.01%	0.004%	0.04%	0.03%
Flooded vegetation	0.1%	3.19%	0.8%	0.06%	1.4%
Crops	8.57%	0.44%	25.3%	31.8%	18.14%
Built area	0%	0.08%	0.09%	1.6%	0.31%
Bare ground	0.2%	4.95%	6.3%	6.14%	5.6%
Rangeland	60.2%	84.56%	59.8%	12.5%	60.32%

## Data Availability

The data presented in this study cannot be shared at this time, as the data form part of an ongoing study.

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
