# Peer review of "Migration Pattern, Habitat Use, and Conservation Status of the Eastern Common Crane (Grus grus lilfordi) from Eastern Mongolia"

_animals, 2023, doi:10.3390/ani13142287_

Round 1

Reviewer 1 Report

The distribution and migratory movements of common cranes Grus g. lilfordi were approximately known in eastern populations. Wintering, breeding, and intermediate migratory stopover areas are roughly delineated. Technological advances and scientific development in the countries deliver field data with shocking space and time accuracies, details unimaginable a few decades ago. These advances allow, for example, the identification of individual migratory strategies in real-time, the learning of juveniles, and the changes associated with their age. By using satellite tagging, this study shows the migratory phenology of five cranes from their first months of life to almost their second year of life (second autumn migration, n= 7 juveniles: CC116, CC119, CC279, CC281, and CC277) and even their third year of life (third autumn migration, juvenile CC116). The work, however, does not explore the effect of age on migratory phenology, despite admitting (Lines 187-191) that the first autumn migration is carried out jointly with their parents. Subsequent migrations are carried out without the guidance and accompaniment of the parents. Individual analyses on the annual changes in migration phenology must be calculated with a sample size of several migration events from n=5 cranes (hint: bird can be defined as a random factor in multivariate statistical analyses). If statistical support for the age effect is not obtained, it should not be concluded that there are no changes in migrations with age (dates, speed, etc.) but rather that this study has not gathered sufficient individuals to study the effect of age in the migration of the common crane.

The work highlights that the eleven cranes spent most of the monitoring time outside the currently protected breeding, wintering, and migratory stopover areas. This observation alone is insufficient to claim an increase in protected areas, both in their extension and location, because the reader is not informed of the number of cranes to which these areas provide protection. It could be that most of the subspecies use the current protected areas. Chance could explain why the eleven marked cranes have not used them. Therefore, the revised text should advise the reader how many cranes of the eastern subpopulation use the protected areas. Given that the size of this subpopulation is known (n= 12,000 – 20,000 cranes, page 2, line 49) and in protected areas, there must be censuses of migratory crane passage, the relevance of the data offered by only eleven cranes would be sufficiently contextualized. Therefore, accurately describing how many cranes are observed in current protected areas must be included in the context of new sites' legal protection. Without this information, it is impossible to assess this work's relevance for preserving the eastern subpopulation of the subspecies G. g. lilfordi. Obviously, the reader must be allowed to know how many cranes use the current protected areas. The manuscript must present the results in their full context: about the protected areas distribution (current manuscript) but also reporting how many of the 12,000 – 20,000 cranes are in those areas and when they are there.

The methods do not explain enough how the individual effect has been treated in the analyses. It is possible to deduce that the sample size in most analyzes is eleven since the statistics used are non-parametric (Mann-Whitney U test or Wilcoxon test, for example). But the results do not indicate the sample sizes in each Figure or Table. Table 1, for example, shows the average departure and arrival dates and the difference between the two. But it should be the median date since there are eleven data. It is an aspect that could be important, given that the difference between the average dates of departure (08/October) and arrival (30/October) in autumn is 22 days, but the duration shown in Table 1 is 49.6 days, more than double. In the case of spring migration, the average duration is compatible with the average start and end dates of that migratory phase. The Mann-Whitney U test values for the difference in migratory duration between fall and spring should be obtained by comparing paired data, one pair per crane. But no information is provided in the Table legend or the main text. It seems evident that these are paired data, but the reader does not know how multi-year data has been treated in this Table. Perhaps the average departure and arrival dates of all migrations for each individual have been calculated to ensure that each crane contributes to the statistical calculation with a data point. Was it like this? Perhaps the means have been calculated with the 36 migrations. It isn't clear enough. I beg your pardon if the main text already explains how the data of 36 migrations were combined within and between individuals is explained in the text and has gone unnoticed while reading.

In common cranes, the size of their area increases with the number of days included in the analyses and when different years are combined. For example, the areas shown in Figures 7 and 8 can change according to the number of data points, but this possibility is not commented on in the manuscript. Moreover: did cranes repeatedly use the same foraging patches between consecutive wintering seasons? Site fidelity decreases with increasing time intervals between daily activity area measurements (Nilsson et al., 2018), suggesting that cranes gradually shift their activity area over time.

Several minor questions must be addressed. Admittedly, some of them could highlight my misunderstanding, so answer them in the letter to the editor.

Minor comments

Page 3, line138. Five years of migrations must include some between-years variation in the mean values in Table 1. In other crane species, migration phenology between years varied due to changes of water levels in stopover sites and air temperature in the breeding grounds (Galtbalt et al., 2022). Please, explain how inter-annual variability was analysed.

Page 4, Table 1. Migration speeds of almost 200 km/day in autumn and 124 km/day in spring are bigger than other populations of Common cranes (45 to 65 km/day; see Ojaste et al., 2020). Please, comment on this considerable difference in the Discussion. Perhaps migration speeds were calculated differently in each study; hence the explanation may be trivial. Maybe Ojaste et al. (2020) called 'migration speed' to 'flight speed'. The manuscript's flight speed is between 54 km/h (autumn) and 48 km/h (spring, page 5, lines 164-169). These figures are alike in both studies. Please, ensure the same words are applied to the same concepts between your study and previous studies.

Page 5, Figure 2. Boxplots mask the inter-bird pattern. For example, the fastest three cranes in autumn can also be the fastest in Spring (Figure 2c). Do the three data points above the Autumn and Spring boxplots belong to the same cranes? The same boxplot can be drawn when (a) all birds change a bit and (b) most birds do not change, but a few birds change a lot. Thus, each pair of points must be linked with a line, and 11 lines must be shown in subplots a-g of Figure 2. Please, replace the boxplots with 11 lines, one per crane. The median of each variable can be added on the left of the autumn points and on the right of the spring points. Notice that even the IQRs can be included on both sides of the eleven lines. Obviously, they must be less wide to leave room to draw eleven lines in the center of each subplot.

Page 6, Figure 3. The procedure to build this Figure is not fully explained in the main text. ¿How many data points were included in each boxplot? Maybe eleven, but the IQR (25%-75%) is small, although some data points highlight flight speeds that may be too fast. For example, a point in autumn (Time = 10h) shows a velocity of 140 km/h. Cranes usually cannot fly that fast.

Some hours could have less than 11 data points in the spring migration because some medians do not show variation. Or maybe all cranes (n=11) fly at the same velocity (granted: very unlikely). Please, provide more details about the results shown in Figure 3.

Figure 3a shows cranes travel at any hour of the 24-h cycle, but common cranes are daytime migrators (e.g., Alonso, J. A. et al., 1990; Alonso, J. C. et al., 1990; Postelnykh et al., 2018). Although these birds occasionally make uninterrupted flights for more than 36 hours (Ojaste et al., 2020; Swanberg, 1986-1987), these events are not frequent enough to stop defining whooping crane migration as basically diurnal. If you agree that the whooping crane is a primarily diurnal migrant, please explain in the Discussion what the nocturnal flight speeds shown in Figure 3 mean. Most, if not all, cranes must be on the ground during the night, but this explanation is not provided in the main text.

Page 12, lines 272-274. Stopover times last longer in the post-breeding migration (autumn) than in the pre-breeding migration (spring), perhaps because young cranes need to feed and drink more than their parents, but also because parents in spring must take over their breeding territories as soon as possible. If you agree, please comment on these elements in the manuscript. If these factors do not determine the migration speed and the number of days in the stopover sites, please explain why in the rebuttal letter. Notice also that Common cranes exhibit territoriality and site fidelity in winter, especially when they carry one or rarely two young  (Alonso et al., 2004). Therefore, successful breeders are expected to travel between breeding and wintering territories. Your sample includes 2 adults with an unknown number of chicks, but the other young may have migrated in the last years with their own chicks. Please, comment on the effects of breeding status (with/without chicks) on migration phenology and to what extent your manuscript has limited scope to contribute to explain this effect. Notice that other factors explaining the seasonal variation (autumn vs. spring) in the migration phenology of birds maybe enrich your Discussion (Schmaljohann, 2018; Schmaljohann et al., 2022)

Page 12, lines 292. It is unclear what "a longer period of time" means. It could be days or years. Satellite GPS/GSM tracking devices last enough years to draw sound conclusions about early life histories by collecting subjects' data from birth to mating (e.g., crane CC116, age: more than three years, Table S2). The success requires fitting several tens of tracking devices because just one out of ten devices, as in the present study, provides three years of data or longer. Therefore, to better understand the migration strategies of Common Cranes, it is necessary to track many more birds (e.g., Postelnykh et al., 2018). Please consider this comment as a way to improve the Discussion, not as a drawback.

       Eight young common cranes were tagged, and their movements were recorded in one crane up to the fourth year (crane CC116). But most of them were recorded for two years, and therefore the study of movements describes the migration of young cranes in their early years of life. In other crane species, birds migrated at different paces based on age and reproductive status, whereas adults with young initiated autumn migration after other birds. For example, adults start the spring migration before subadult birds (Whooping Crane, see Pearse et al., 2020). Likely the common cranes show the same age pattern of migration. Therefore, your results mainly show the migration phenology of young and immature birds, not adult birds. Comment on this possible bias of your bird sample and consider updating the title of the manuscript.

Page 12, lines 313-321. Establishing and expanding protected areas do not ensure farmers scare and drive away the common cranes. To achieve a more balanced management plan is essential an effective combination of multiple management practices (Pekarsky et al., 2021).

Typos

Page 1, line 17. '2017 to 2019' were the capture and tagging years, but 36 migrations were collected between 2019 and 2021. Therefore, this sentence must end with '2017 to 2021.'

Page 11, Figure 8. Check country boundaries because some colors do not match the legend. Idem Figure S1.

 Page 12, line 281. The reference of Liu et al. (2018) may be not [35] but [38]. Revise the citations numeration in the main text.

 Page 12, line 284. Check citations [29, 30] because they are not migration studies of Common Cranes. Mensson and Namalainen (2012) described swans' migration, and Nilsson et al. (2013) compared spring vs autumn migration seasons, but not in common cranes. Perhaps the references must be [34, 35].

Page 14, line 403. Check this expression: ):48

References

Please, be aware that this list of references is illustrative and not mandatory. None of them needs to be included in the updated manuscript.

Alonso JA, Alonso JC, Cantos FJ, and Bautista LM. 1990. Spring crane Grus grus migration through Gallocanta, Spain. 2. Timing and pattern of daily departures. Ardea 78:379-386. https://digital.csic.es/bitstream/10261/43954/1/Alonso%20Ardea78%20p379%201990.pdf

Alonso JC, Alonso JA, Cantos FJ, and Bautista LM. 1990. Spring crane Grus grus migration through Gallocanta, Spain. 1. Daily variations in migration volume. Ardea 78:365-378. https://digital.csic.es/bitstream/10261/43953/1/Alonso%20Ardea78%20p365%201990.pdf

Alonso JC, Bautista LM, and Alonso JA. 2004. Family-based territoriality vs flocking in wintering common cranes Grus grus. Journal of Avian Biology 35:434-444. https://www.researchgate.net/publication/230382237

Galtbalt B, Natsagdorj T, Sukhbaatar T, Mirande C, Archibald G, Batbayar N, and Klaassen M. 2022. Breeding and migration performance metrics highlight challenges for White-naped Cranes. Scientific Reports 12:13. https://doi.org/10.1038/s41598-022-23108-w

Nilsson L, Aronsson M, Persson J, and Månsson J. 2018. Drifting space use of common cranes—Is there a mismatch between daytime behaviour and management? Ecological Indicators 85:556-562. https://doi.org/10.1016/j.ecolind.2017.11.007

Ojaste I, Leito A, Suorsa P, Hedenstrom A, Sepp K, Leivits M, Sellis U, and Vali U. 2020. From northern Europe to Ethiopia: long-distance migration of Common Cranes (Grus grus). Ornis Fennica 97:12-25. https://www.ornisfennica.org/pdf/latest/19Ojaste.pdf

Pearse AT, Metzger KL, Brandt DA, Bidwell MT, Harner MJ, Baasch DM, and Harrell W. 2020. Heterogeneity in migration strategies of Whooping Cranes. Condor 122:1-15. https://doi.org/10.1093/condor/duz056

Pekarsky S, Schiffner I, Markin Y, and Nathan R. 2021. Using movement ecology to evaluate the effectiveness of multiple human-wildlife conflict management practices. Biological Conservation 262:11. https://doi.org/10.1016/j.biocon.2021.109306

Postelnykh KA, Markin YM, Pekarsky S, and Nathan R. 2021. Movements patterns of first-year juvenile Eurasian cranes revealed using GPS-transmitter data. Paper presented at the European Crane Conference, 2018. (P Dulau, ed. eds.). Arjuzanx, France. Conference organized by Syndicat Mixte de Gestion des Milieux Naturels – Réserve Nationale de Faune Sauvage d’Arjuzanx. Proceedings of the European Crane Conference 9: 188. https://doi.org/. https://www.researchgate.net/profile/Dessalegn-Gemeda/publication/357573888_actes_de_la_conference/links/61d49073b8305f7c4b20af82/actes-de-la-conference.pdf#page=49

Schmaljohann H. 2018. Proximate mechanisms affecting seasonal differences in migration speed of avian species. Scientific Reports 8:4106. https://doi.org/10.1038/s41598-018-22421-7

Schmaljohann H, Eikenaar C, and Sapir N. 2022. Understanding the ecological and evolutionary function of stopover in migrating birds. Biological Reviews in press:22. https://doi.org/10.1111/brv.12839

Swanberg PO. 1986-1987. Studies on the influence of weather on migrating cranes (Grus grus) in Sweden. Aquila 93-94:203-212. http://epa.niif.hu/01600/01603/00075/pdf/Aquila_EPA-01603_1986-1987_203-212.pdf

Author Response

Response to Reviewer 

Title: Migration Pattern, Habitat Use, and Conservation Status of the Eastern Common Crane (Grus grus lilfordi) from Eastern Mongolia

Manuscript ID: 2430220

Dear Reviewer,

We sincerely thank you for your precious time in reviewing our manuscript and providing valuable comments. It was the reviewers' valuable comments and suggestions that led to possible improvements in the current version. The authors have carefully considered the comments and made a concerted effort to adequately respond to each suggestion received from the reviewers. The authors welcome further constructive comments if any.

Below we provide the point-by-point responses. All modifications in the manuscript have been tracked changed and highlighted in red. We hope that reviewers find this manuscript acceptable for publication in the Animal Journal.

Best regards,

Baasansuren Erdenechimeg

baaskaa.1227@gmail.com

School of Ecology and Nature Conservation, Beijing Forestry University, Beijing, China.

Mongolian Bird Conservation Center, Ulaanbaatar, Mongolia.

Response to Reviewer 1

[Comment and suggestions 1] The distribution and migratory movements of common cranes Grus g. lilfordi were approximately known in eastern populations. Wintering, breeding, and intermediate migratory stopover areas are roughly delineated. Technological advances and scientific development in the countries deliver field data with shocking space and time accuracies, details unimaginable a few decades ago. These advances allow, for example, the identification of individual migratory strategies in real-time, the learning of juveniles, and the changes associated with their age. By using satellite tagging, this study shows the migratory phenology of five cranes from their first months of life to almost their second year of life (second autumn migration, n=7 juveniles: CC116, CC119, CC279, CC281, and CC277) and even their third year of life (third autumn migration, juvenile CC116). The work, however, does not explore the effect of age on migratory phenology, despite admitting (Lines 187-191) that the first autumn migration is carried out jointly with their parents. Subsequent migrations are carried out without the guidance and accompaniment of the parents. Individual analyses on the annual changes in migration phenology must be calculated with a sample size of several migration events from n=5 cranes (hint: bird can be defined as a random factor in multivariate statistical analyses). If statistical support for the age effect is not obtained, it should not be concluded that there are no changes in migrations with age (dates, speed, etc.) but rather that this study has not gathered sufficient individuals to study the effect of age in the migration of the common crane.

Response: Thank you for your review on our manuscript. In this study, we used a total of 8 juvenile birds (first autumn migration n=8, second autumn migration n=5, third autumn migration n=3, fourth autumn migration n=1) and 3 adult birds (first autumn migration n=3, three adult birds spring migration started earlier than juvenile birds, but the migration data was interrupted on the way because they could not reach the breeding areas) migration data. Migration data collected from juvenile and adult birds differed, and age-specific migration patterns were not analyzed because the data collected was insufficient to analyze age-specific migration patterns.

[Comment and suggestions 2] The work highlights that the eleven cranes spent most of the monitoring time outside the currently protected breeding, wintering, and migratory stopover areas. This observation alone is insufficient to claim an increase in protected areas, both in their extension and location, because the reader is not informed of the number of cranes to which these areas provide protection. It could be that most of the subspecies use the current protected areas. Chance could explain why the eleven marked cranes have not used them. Therefore, the revised text should advise the reader how many cranes of the eastern subpopulation use the protected areas. Given that the size of this subpopulation is known (n=12,000 – 20,000 cranes, page 2, line 49) and in protected areas, there must be censuses of migratory crane passage, the relevance of the data offered by only eleven cranes would be sufficiently contextualized. Therefore, accurately describing how many cranes are observed in current protected areas must be included in the context of new sites' legal protection. Without this information, it is impossible to assess this work's relevance for preserving the eastern subpopulation of the subspecies G. g. lilfordi. Obviously, the reader must be allowed to know how many cranes use the current protected areas. The manuscript must present the results in their full context: about the protected areas distribution (current manuscript) but also reporting how many of the 12,000 – 20,000 cranes are in those areas and when they are there.

Response: Thank you for your comment. The comment was really relevant and this information was added to the results. You can see added information [Page 12, Line 302-305].

[Comment and suggestions 3] The methods do not explain enough how the individual effect has been treated in the analyses. It is possible to deduce that the sample size in most analyzes is eleven since the statistics used are non-parametric (Mann-Whitney U test or Wilcoxon test, for example). But the results do not indicate the sample sizes in each Figure or Table. Table 1, for example, shows the average departure and arrival dates and the difference between the two. But it should be the median date since there are eleven data. It is an aspect that could be important, given that the difference between the average dates of departure (08/October) and arrival (30/October) in autumn is 22 days, but the duration shown in Table 1 is 49.6 days, more than double. In the case of spring migration, the average duration is compatible with the average start and end dates of that migratory phase. The Mann-Whitney U test values for the difference in migratory duration between fall and spring should be obtained by comparing paired data, one pair per crane. But no information is provided in the Table legend or the main text. It seems evident that these are paired data, but the reader does not know how multi-year data has been treated in this Table. Perhaps the average departure and arrival dates of all migrations for each individual have been calculated to ensure that each crane contributes to the statistical calculation with a data point. Was it like this? Perhaps the means have been calculated with the 36 migrations. It isn't clear enough. I beg your pardon if the main text already explains how the data of 36 migrations were combined within and between individuals is explained in the text and has gone unnoticed while reading. 

Response: Thank you for your question. Yes, the average departure and arrival dates of all 36 migrations for each individual have been calculated and shown in Table 1. For example: In table 1, the average departure was 8/X±44 in autumn. Because, the earliest departure was 25 Aug, and the latest was 22 Nov. We have added to the methods section how we calculated the average departure and arrival dates during the migration [Page 3, Line 103-104].

[Comment and suggestions 4] In common cranes, the size of their area increases with the number of days included in the analyses and when different years are combined. For example, the areas shown in Figures 7 and 8 can change according to the number of data points, but this possibility is not commented on in the manuscript. Moreover: did cranes repeatedly use the same foraging patches between consecutive wintering seasons? Site fidelity decreases with increasing time intervals between daily activity area measurements (Nilsson et al., 2018), suggesting that cranes gradually shift their activity area over time.

Several minor questions must be addressed. Admittedly, some of them could highlight my misunderstanding, so answer them in the letter to the editor.

Response: Thank you very much for pointing this out. Although we agree with your comment, we would like to make a few responses. We found that tracked five individuals (CC116, CC277, CC119, CC279, CC281) in Eastern Mongolia for which we have more than one year of tracking data showed strong site-fidelity during both wintering and breeding seasons. For example, the crane CC116 was captured in Chukh Lake (Northeastern Mongolia) in 2017. When we catch, it was a juvenile. From this individual, we collected full migration trips (4 autumn and 4 spring migrations) from 25 Jul 2017 to 6 Apr 2021. Because the crane CC116 was a young bird and it was spending the first and second years of summer as a wonder. We think this individual was looking for a breeding habitat and pair with other young birds. Although, CC116 was vagrant in the summer, but it returned to its parent's nesting territory (Chukh Lake) in April 2018 and 2019. The CC116 was wintered in Luan River, Tangshan, Hebei, China in 2018 and 2019. As well as wintered Yellow River Delta, Shandong, China in 2020 and 2021. These transmitter data indicate that the cranes are site-fidelity.

If we collect long-term tracking data of the cranes, we can combine them and will more accurately and clearly determine the areas used by tracked cranes. By combining long-term data, we can determine which areas are most important and develop conservation recommendations. In addition to our tracked cranes, we think many cranes without transmitters are using these areas (Figures 7 and 8).

[Minor comments 1] Page 3, line138. Five years of migrations must include some between-years variation in the mean values in Table 1. In other crane species, migration phenology between years varied due to changes of water levels in stopover sites and air temperature in the breeding grounds (Galtbalt et al., 2022). Please, explain how inter-annual variability was analysed.

Response: We thank you for your comment. Although we hypothesized that migration phenology (migration duration, departure, and arrival date, etc.) between years varied due to changes of water levels in stopover sites, the annual year's weather, food availability, fledgling age, and habitat suitability in our study, but we cannot say for sure what exactly caused the migration phenology between years to be different, so this did not explain. Also, we did not analyze the differences in annual migration phenology in detail. We think the reader can find a summary of the annual migration patterns of each individual in Supplementary material, Table S1.

In this study, we aimed to determine the migration patterns of this subspecies and to provide basic information about migration patterns. In the future, we will study age-specific migration patterns and annual migration phenology differences in more detail.

[Minor comments 2] Page 4, Table 1. Migration speeds of almost 200 km/day in autumn and 124 km/day in spring are bigger than other populations of Common cranes (45 to 65 km/day; see Ojaste et al., 2020). Please, comment on this considerable difference in the Discussion. Perhaps migration speeds were calculated differently in each study; hence the explanation may be trivial. Maybe Ojaste et al. (2020) called 'migration speed' to 'flight speed'. The manuscript's flight speed is between 54 km/h (autumn) and 48 km/h (spring, page 5, lines 164-169). These figures are alike in both studies. Please, ensure the same words are applied to the same concepts between your study and previous studies.

Response: We would like to thank the reviewer for having suggested this important point. We carefully read the Ojaste et al. (2020) manuscript again and compared the migration speed with our study result. In our study, we defined migration speed according to the following definition.

The migration speed was calculated by dividing the total migration distance (in kilometers) by the total migration duration (in days) (Chen and Dobra, 2020). You can see [Page 3, Line 116-117].

In the study by Ojastei et al, the overall migration speeds of the Finnish and Estonian cranes were 64.7 ± 31.8 km/day (n=7) and 44.7 ± 11.3 km/day (n=11). The total distance was from 3,520–6,527 km and 2,177–5,862 km for the Finnish and Estonian cranes. The researchers highlighted in their manuscript the shortest migration time of 32 days covering 6,178 km. We just tried to calculate the migration distance using this result according to the methodology described above, and the migration distance was 192 km/h. So, we think Ojaste et al. (2020) called 'flight speed' to migration speed'. But they did not mention in the methodology section of their manuscript how they calculated the migration speed.

[Minor comments 3] Page 5, Figure 2. Boxplots mask the inter-bird pattern. For example, the fastest three cranes in autumn can also be the fastest in Spring (Figure 2c). Do the three data points above the Autumn and Spring boxplots belong to the same cranes? The same boxplot can be drawn when (a) all birds change a bit and (b) most birds do not change, but a few birds change a lot. Thus, each pair of points must be linked with a line, and 11 lines must be shown in subplots a-g of Figure 2. Please, replace the boxplots with 11 lines, one per crane. The median of each variable can be added on the left of the autumn points and on the right of the spring points. Notice that even the IQRs can be included on both sides of the eleven lines. Obviously, they must be less wide to leave room to draw eleven lines in the center of each subplot.
Response: First, we thank you for your important comment. You are right. We tried to replace the boxplots with 11 lines, one per crane. But our data was not enough for subplots. Because we checked our data again. Some fastest cranes (CC277, CC279, and CC281) in autumn were also fast in spring. But some cranes (CC119) only were fast in spring, normal during the autumn [Figure 2c and Table S2]. But some (CC210) cranes' spring migration data was interrupted, so it was not possible to show each crane's migration patterns on subplots. Although we couldn't make subplots, Table S2 shows each crane's migration pattern in autumn and spring details.

We believe that it will be easier for readers to understand this boxplot (migration patterns of the combined data from 11 individuals) in Figure 2 a-g.

[Minor comments 4] Page 6, Figure 3. The procedure to build this Figure is not fully explained in the main text. How many data points were included in each boxplot? Maybe eleven, but the IQR (25%-75%) is small, although some data points highlight flight speeds that may be too fast. For example, a point in autumn (Time = 10h) shows a velocity of 140 km/h. Cranes usually cannot fly that fast.

Response: Thank you for your question. According to your suggestion, we explained the procedure to build this figure [Page 3, Line 145-146]. A total of 15-64 data points were included in each boxplot in autumn and 9-58 data points in spring.

We checked our data again. During autumn migration, flight speed averaged 53.9±22.8 km/h, ranging from 10.2 to 144.7 km/h. During spring migration, the flight speed averaged 48.5±21.48 km/h, ranging from 10.5 to 145.3 km/h.

Filtering data by the times that flew at a speed of over 100 km/h during the migration, it was between 00:00 to 21:00 pm in (n=8) autumn and 00:00 to 17:00 pm in spring (n=4). Some (CC061, CC116, and CC201) individuals were migrated in the night at 12-4 pm.

[Minor comments 5] Some hours could have less than 11 data points in the spring migration because some medians do not show variation. Or maybe all cranes (n=11) fly at the same velocity (granted: very unlikely). Please, provide more details about the results shown in Figure 3.

Response: Thank you for pointing this out. The autumn migration data was larger than the spring migration data. Because some individuals' (CC209, CC210, and CC213) spring migration data was interrupted. So, in this analysis, spring migration data was not enough.

[Minor comments 6] Figure 3a shows cranes travel at any hour of the 24-h cycle, but common cranes are daytime migrators (e.g., Alonso, J. A. et al., 1990; Alonso, J. C. et al., 1990; Postelnykh et al., 2018). Although these birds occasionally make uninterrupted flights for more than 36 hours (Ojaste et al., 2020; Swanberg, 1986-1987), these events are not frequent enough to stop defining whooping crane migration as basically diurnal. If you agree that the whooping crane is a primarily diurnal migrant, please explain in the Discussion what the nocturnal flight speeds shown in Figure 3 mean. Most, if not all, cranes must be on the ground during the night, but this explanation is not provided in the main text.

Response: We thank you for your important suggestion. Revised accordingly [Page 6, Line 227-228; Page 13, Line 361-367]. We read and used these citations in the discussion. Thanks for reading many manuscripts and suggesting important references for us.

[Minor comments 7] Page 12, lines 272-274. Stopover times last longer in the post-breeding migration (autumn) than in the pre-breeding migration (spring), perhaps because young cranes need to feed and drink more than their parents, but also because parents in spring must take over their breeding territories as soon as possible. If you agree, please comment on these elements in the manuscript. If these factors do not determine the migration speed and the number of days in the stopover sites, please explain why in the rebuttal letter. Notice also that Common cranes exhibit territoriality and site fidelity in winter, especially when they carry one or rarely two young (Alonso et al., 2004). Therefore, successful breeders are expected to travel between breeding and wintering territories. Your sample includes 2 adults with an unknown number of chicks, but the other young may have migrated in the last years with their own chicks. Please, comment on the effects of breeding status (with/without chicks) on migration phenology and to what extent your manuscript has limited scope to contribute to explain this effect. Notice that other factors explaining the seasonal variation (autumn vs. spring) in the migration phenology of birds maybe enrich your Discussion (Schmaljohann, 2018; Schmaljohann et al., 2022).

Response: Thank you very much for your valuable suggestion. We agree with you. Revised accordingly [Page 13, Line 356-360].

Our tracked (CC209, CC210, CC213) adults were without chicks.

[Minor comments 8] Page 12, lines 292. It is unclear what "a longer period of time" means. It could be days or years. Satellite GPS/GSM tracking devices last enough years to draw sound conclusions about early life histories by collecting subjects' data from birth to mating (e.g., crane CC116, age: more than three years, Table S2). The success requires fitting several tens of tracking devices because just one out of ten devices, as in the present study, provides three years of data or longer. Therefore, to better understand the migration strategies of Common Cranes, it is necessary to track many more birds (e.g., Postelnykh et al., 2018). Please consider this comment as a way to improve the Discussion, not as a drawback.

Response: Thank you for your important suggestion. According to your suggestion, we added this paragraph to the discussion [Page 14, Line 379-382].

[Minor comments 9] Eight young common cranes were tagged, and their movements were recorded in one crane up to the fourth year (crane CC116). But most of them were recorded for two years, and therefore the study of movements describes the migration of young cranes in their early years of life. In other crane species, birds migrated at different paces based on age and reproductive status, whereas adults with young initiated autumn migration after other birds. For example, adults start the spring migration before subadult birds (Whooping Crane, see Pearse et al., 2020). Likely the common cranes show the same age pattern of migration. Therefore, your results mainly show the migration phenology of young and immature birds, not adult birds. Comment on this possible bias of your bird sample and consider updating the title of the manuscript.

Response: Thank you very much for your suggestion. We agree with you. Our transmitter data showed that cranes of breeding age (CC116, fourth year) and adults (n=3, but spring migration data were interrupted on the way) started their spring migration earlier than young birds. But this data is not sufficient to study differences in age-specific migration patterns.

We've discussed re-naming our manuscript before. Most of the cranes were juveniles. But in this analysis, we used three adults (without chick) and subadults migration data. Therefore, it was considered less appropriate to rename the manuscript because our study used data on the migration of cranes of a certain age.

[Minor comments 10] Page 12, lines 313-321. Establishing and expanding protected areas do not ensure farmers scare and drive away the common cranes. To achieve a more balanced management plan is essential an effective combination of multiple management practices (Pekarsky et al., 2021).

References

Please, be aware that this list of references is illustrative and not mandatory. None of them needs to be included in the updated manuscript.

Alonso JA, Alonso JC, Cantos FJ, and Bautista LM. 1990. Spring crane Grus grus migration through Gallocanta, Spain. 2. Timing and pattern of daily departures. Ardea 78:379-386. https://digital.csic.es/bitstream/10261/43954/1/Alonso%20Ardea78%20p379%201990.pdf

Alonso JC, Alonso JA, Cantos FJ, and Bautista LM. 1990. Spring crane Grus grus migration through Gallocanta, Spain. 1. Daily variations in migration volume. Ardea 78:365-378. https://digital.csic.es/bitstream/10261/43953/1/Alonso%20Ardea78%20p365%201990.pdf

Alonso JC, Bautista LM, and Alonso JA. 2004. Family-based territoriality vs flocking in wintering common cranes Grus grus. Journal of Avian Biology 35:434-444. https://www.researchgate.net/publication/230382237

Galtbalt B, Natsagdorj T, Sukhbaatar T, Mirande C, Archibald G, Batbayar N, and Klaassen M. 2022. Breeding and migration performance metrics highlight challenges for White-naped Cranes. Scientific Reports 12:13. https://doi.org/10.1038/s41598-022-23108-w

Nilsson L, Aronsson M, Persson J, and Månsson J. 2018. Drifting space use of common cranes—Is there a mismatch between daytime behaviour and management? Ecological Indicators 85:556-562. https://doi.org/10.1016/j.ecolind.2017.11.007

Ojaste I, Leito A, Suorsa P, Hedenstrom A, Sepp K, Leivits M, Sellis U, and Vali U. 2020. From northern Europe to Ethiopia: long-distance migration of Common Cranes (Grus grus). Ornis Fennica 97:12-25. https://www.ornisfennica.org/pdf/latest/19Ojaste.pdf

Pearse AT, Metzger KL, Brandt DA, Bidwell MT, Harner MJ, Baasch DM, and Harrell W. 2020. Heterogeneity in migration strategies of Whooping Cranes. Condor 122:1-15. https://doi.org/10.1093/condor/duz056

Pekarsky S, Schiffner I, Markin Y, and Nathan R. 2021. Using movement ecology to evaluate the effectiveness of multiple human-wildlife conflict management practices. Biological Conservation 262:11. https://doi.org/10.1016/j.biocon.2021.109306

Postelnykh KA, Markin YM, Pekarsky S, and Nathan R. 2021. Movements patterns of first-year juvenile Eurasian cranes revealed using GPS-transmitter data. Paper presented at the European Crane Conference, 2018. (P Dulau, ed. eds.). Arjuzanx, France. Conference organized by Syndicat Mixte de Gestion des Milieux Naturels – Réserve Nationale de Faune Sauvage d’Arjuzanx. Proceedings of the European Crane Conference 9: 188. https://doi.org/. https://www.researchgate.net/profile/Dessalegn-Gemeda/publication/357573888_actes_de_la_conference/links/61d49073b8305f7c4b20af82/actes-de-la-conference.pdf#page=49

Schmaljohann H. 2018. Proximate mechanisms affecting seasonal differences in migration speed of avian species. Scientific Reports 8:4106. https://doi.org/10.1038/s41598-018-22421-7

Schmaljohann H, Eikenaar C, and Sapir N. 2022. Understanding the ecological and evolutionary function of stopover in migrating birds. Biological Reviews in press:22. https://doi.org/10.1111/brv.12839

Swanberg PO. 1986-1987. Studies on the influence of weather on migrating cranes (Grus grus) in Sweden. Aquila 93-94:203-212. http://epa.niif.hu/01600/01603/00075/pdf/Aquila_EPA-01603_1986-1987_203-212.pdf

Response: Thank you very much for your comment. Revised accordingly [Page 14, Line 409-411]. We read and used some references. Thanks for suggesting important references for us.

[Typos 1] Page 1, line 17. '2017 to 2019' were the capture and tagging years, but 36 migrations were collected between 2019 and 2021. Therefore, this sentence must end with '2017 to 2021.

Response: We thank the reviewer for having suggested this important typo. We revised the sentence as follows:

Using GPS/GSM tracking data, 36 complete migrations of 11 individuals were obtained from 2017 to 2021 [Page 1, Line 21].

[Typos 2] Page 11, Figure 8. Check country boundaries because some colors do not match the legend. Idem Figure S1.

Response: Thank you very much for your nice reminder. We revised most of the figure's country boundary color. The color matched the legend. [Page 8, Figure 5; Page 9, Figure 6; Page 11, Figure 8 and Figure S1].

[Typos 3] Page 12, line 281. The reference of Liu et al. (2018) may be not [35] but [38]. Revise the citations numeration in the main text.

Response: Thank you for your reminder. We checked that all references matched the citation numeration of the manuscript. Revised the citation's numeration in the main text. The reference of Liu et al. (2018) it is [36]. You can see [Page 13, Line 346]. Because, we have made some mistakes and forgot to delete references [29, 30, 31]. So, we deleted those references [Page 16, Line 504-509]. 

[Typos 4] Page 12, line 284. Check citations [29, 30] because they are not migration studies of Common Cranes. Mensson and Namalainen (2012) described swans' migration, and Nilsson et al. (2013) compared spring vs autumn migration seasons, but not in common cranes. Perhaps the references must be [34, 35].

Response: We thank you for your reminder. These [29, 30] citations were revised. You can see [Page 132, Line 350]. We have forgotten to delete some references [29, 30, 31]. Therefore, the order of citations and references was different. [30, 31] these citations are surveys of Ojaste et al. (2020), and Ilyashenko et al. (2023). You can find it [Page 16, Line 510-513]. 

[Typos 5] Page 14, line 403. Check this expression: ):48

Response: Thanks for your reminder. We deleted [29] this reference [Page 16, Line 504-505].

Reviewer 2 Report

The MS ‘Migration pattern, habitat use, and conservation status of the eastern Common Crane (Grus grus lilfordi) from eastern Mongolia reports the results of a satellite tracking study of 11 common cranes from their breeding grounds in eastern Mongolia to wintering grounds in eastern China. This is a sound study, based on a sufficient sample. However, I do have several points to raise.

Line 32 ‘allowing people’ – it would be better to write ‘allowing researchers’

Line 39: ‘Currently… four subspecies… are recognized’, not ‘identified’.

Line 45: should be ‘Population size of the eastern subspecies is estimated…’

When mentioning common crane migration in the Amudarya river valley in Central Asia, you may want to make a reference to the following paper:

Bulyuk V.N., Shamuradov A.K. 1995. The migration of the Common Crane, Grus grus, in southern Turkmenistan. Zoology in the Middle East 11: 21-30. https://doi.org/10.1080/09397140.1995.10637667

Lines 95-96: When you analyze satellite tracking data, you cannot identify stopover sites ‘as places where birds rest and feed for more than two days’. From the tracks, you do not know what they are doing, you can only speculate. You should identify stopovers sites as places where birds remain stationary for e.g. >2 days.

Line 102: you first mention KDE maps, and only later explain what KDE stands for. You should explain it at the first mention immediately. At line 117, you explain it again. This should only be done once.

Line 177, legend to Figure 3. When you compare flight altitudes between different portions of the migratory route, you should provide significance level or mention that the difference was not significant (what is the point reporting it then?).

In the Discussion section, I can see no point comparing the length of migratory routes between different common crane populations and other crane species. Different populations have migratory routes governed by the location of their respective breeding and wintering quarters. Some avian population migrate longer distances, others shorter distances. Unless you find a pattern, this comparison is trivial.

English usage is generally okay; however, I have several suggestions mentioned earlier.

Author Response

Response to Reviewers

[Cover Letter]

Title: Migration Pattern, Habitat Use, and Conservation Status of the Eastern Common Crane (Grus grus lilfordi) from Eastern Mongolia

Manuscript ID: 2430220

Dear Reviewer,

We sincerely thank you for your precious time in reviewing our manuscript and providing valuable comments. It was the reviewers' valuable comments and suggestions that led to possible improvements in the current version. The authors have carefully considered the comments and made a concerted effort to adequately respond to each suggestion received from the reviewers. The authors welcome further constructive comments if any.

Below we provide the point-by-point responses. All modifications in the manuscript have been tracked changed and highlighted in red. We hope that reviewers find this manuscript acceptable for publication in the Animal Journal.

Best regards,

Baasansuren Erdenechimeg

baaskaa.1227@gmail.com

School of Ecology and Nature Conservation, Beijing Forestry University, Beijing, China.

Mongolian Bird Conservation Center, Ulaanbaatar, Mongolia.

Response to Reviewer 2

[Comment and suggestions 1] The MS ‘Migration pattern, habitat use, and conservation status of the eastern Common Crane (Grus grus lilfordi) from eastern Mongolia reports the results of a satellite tracking study of 11 common cranes from their breeding grounds in eastern Mongolia to wintering grounds in eastern China. This is a sound study, based on a sufficient sample. However, I do have several points to raise.

Response: We thank you for your review and suggestions in our manuscript.

[Comment and suggestions 2] Line 32 ‘allowing people – it would be better to write ‘allowing researchers’

Response: Thank you for your comment. According to your comment, we revised this sentence as follows:

In the early 1990s, with the continuous development of science and technology, the research methods of bird migration also made significant progress, allowing researchers to better understand the migratory behavior of birds [Page 1, Line 36-37].

[Comment and suggestions 3] Line 39: ‘Currently… four subspecies… are recognized’, not ‘identified’.

Response: Thank you. We revised ‘identified’ to ‘recognized’ [Page 2, Line 44].

[Comment and suggestions 4] Line 45: should be ‘Population size of the eastern subspecies is estimated…’

Response: Thank you for the comment. Revised accordingly [Page 2, Line 49].

[Comment and suggestions 5] When mentioning common crane migration in the Amudarya river valley in Central Asia, you may want to make a reference to the following paper:

Bulyuk V.N., Shamuradov A.K. 1995. The migration of the Common Crane, Grus grus, in southern Turkmenistan. Zoology in the Middle East 11: 21-30. https://doi.org/10.1080/09397140.1995.10637667

Response: We thank you for your suggestion. Added and cited this reference. You can see [Page 2, Line 52; Page 15, Line 470-741].

[Comment and suggestions 6] Lines 95-96: When you analyze satellite tracking data, you cannot identify stopover sites ‘as places where birds rest and feed for more than two days. From the tracks, you do not know what they are doing, you can only speculate. You should identify stopovers sites as places where birds remain stationary for e.g. >2 days.

Response: We would like to thank the reviewer for having suggested this important point. Revised accordingly [Page 3, Line 124].

[Comment and suggestions 7] Line 102: you first mention KDE maps, and only later explain what KDE stands for. You should explain it at the first mention immediately. At line 117, you explain it again. This should only be done once.

Response: Thank you. Revised accordingly [Page 4, Line 157-160].

[Comment and suggestions 8] Line 177, legend to Figure 3. When you compare flight altitudes between different portions of the migratory route, you should provide significance level or mention that the difference was not significant (what is the point reporting it then?).

Response: Thank you very much for your suggestion. We think you are asking about Figure 4. There was a difference in flight altitude between crossing Khingan mountain and other places (Wilcoxon test, w=276, 478; p<0.05). Also, you can see the result [Page 7-8, Line 232-241] and discussion [Page 13, Line 340-345].

[Comment and suggestions 9] In the Discussion section, I can see no point comparing the length of migratory routes between different common crane populations and other crane species. Different populations have migratory routes governed by the location of their respective breeding and wintering quarters. Some avian population migrate longer distances, others shorter distances. Unless you find a pattern, this comparison is trivial.

Response: We thank you for your important suggestion. We agree with you. Although the overall migration patterns of the Common Crane populations were relatively similar, there were significant differences in White-naped Cranes migrated from the exact same location as the Eastern Common Cranes. You can see [Page 13, Line 325-329].

[Comment and suggestions 10] Comments on the Quality of English Language

English usage is generally okay; however, I have several suggestions mentioned earlier.

Response: Thank you. Our friend Leigh-Ann Barran helped us to review our manuscript in native English.
